# Imaging the field inside nanophotonic accelerators

Tal Fishman[1,4] ✉, Urs Haeusler [2,3,4], Raphael Dahan[1], Michael Yannai[1], Yuval Adiv[1], Tom Lenkiewicz Abudi[1], Roy Shiloh[2], Ori Eyal[1], Peyman Yousefi[2], Gadi Eisenstein[1], Peter Hommelhoff [2] & Ido Kaminer [1] ✉

Controlling optical fields on the subwavelength scale is at the core of nanophotonics. Laser-driven nanophotonic particle accelerators promise a compact alternative to conventional radiofrequency-based accelerators. Efficient electron acceleration in nanophotonic devices critically depends on achieving nanometer control of the internal optical nearfield. However, these nearfields have so far been inaccessible due to the complexity of the devices and their geometrical constraints, hampering the design of future nanophotonic accelerators. Here we image the field distribution inside a nanophotonic accelerator, for which we developed a technique for frequency-tunable deep-subwavelength resolution of nearfields based on photon-induced nearfield electron-microscopy. Our experiments, complemented by 3D simulations, unveil surprising deviations in two leading nanophotonic accelerator designs, showing complex field distributions related to intricate 3D features in the device and its fabrication tolerances. We envision an extension of our method for full 3D field tomography, which is key for the future design of highly efficient nanophotonic devices.

Particle accelerators are critical components in modern industrial, academic and medical infrastructure. Their applications range from X-ray generation for cancer treatment and medical imaging to the high-end experiments of high energy physics[1–3]. Current accelerators are based on radiofrequency technology that makes them large and expensive. Shrinking the size of accelerators could open new possibilities for numerous applications, including electron diffraction and microscopy, portable medical X-ray sources, and coherent probes for quantum information science[4,5].

One approach toward compact particle accelerators facilitates efficient interaction between the particles and light by using short laser pulses and nanophotonic devices built from dielectric materials[6]. An important advantage of these nanophotonic dielectric laser acceleration (DLA) devices, compared to metal or superconducting radiofrequency accelerators, lies in the two order of magnitude higher damage threshold of dielectrics at optical frequencies[7]. Together with

their small footprint on the order of millimeters and the use of off-the-shelf infrared laser sources as the energy source for acceleration, DLA promises an economical and compact alternative to radiofrequency accelerators. The operation of DLA devices is based on the inverse Smith–Purcell effect[8–10], a process of stimulated absorption and emission of photons by an electron that passes close to a periodic structure. Over the last sixty years, various methods for enhancing the fundamental electron-light interaction in DLA structures have been proposed[6,11–18]. The first successful DLA demonstrations from 2013 showed a few dozens of MeV/m acceleration gradient for sub-relativistic electrons[8,19] and up to 250 MeV/m for relativistic electrons[20]. Since then, continuous improvements in DLA designs and laser coupling efficiency, benefitting from the mature technology of silicon photonics, led to the demonstration of acceleration gradients up to and above the GeV/m scale[17,21–27]. The current bottleneck for pushing forward DLA technology is in the precise control of the

[1]Department of Electrical and Computer Engineering, Technion – Israel Institute of Technology, Haifa 32000, Israel. [2]Department of Physics, Friedrich-Alexander-Universität Erlangen-Nürnberg (FAU), Staudtstraße 1, Erlangen 91058, Germany. [3]Cavendish Laboratory, University of Cambridge, JJ Thomson Avenue, Cambridge CB3 0HE, UK. [4]These authors contributed equally: Tal Fishman, Urs Haeusler. ✉e-mail: ftal@technion.ac.il; kaminer@technion.ac.il

electron trajectories, which depend on the exact nearfield distribution inside the DLA. This high degree of control is necessary for longer and more modular acceleration structures and for the concatenation of multiple such devices.

Further optimization and advanced electron beam focusing techniques, such as alternating phase focusing[18,28,29], require a detailed understanding of the 3D nearfield distribution inside the accelerator structure. While simulations can provide extensive insight into the field distribution, they depend on the precise 3D knowledge of the fabricated structure dimensions and laser beam parameters, which are rarely fully accessible. Consequently, direct characterization of the field distribution is of paramount importance. However, until now, no experimental method has been able to measure the actual field distribution inside nanophotonic accelerators.

Here, we demonstrate the deep-subwavelength characterization of the accelerating electric field inside a DLA channel. To image the nearfield, we modify a transmission electron microscope (TEM) such that electrons first pass through the DLA channel and are then filtered

by their energy – a technique known as energy-filtered transmission electron microscopy (EFTEM, see Fig. 1a). By collecting only the electrons that gained energy from their interaction with the field, we obtain an image of the accelerating field. The field strength across the channel can be reconstructed from the density of electrons at different positions after the energy filtering.

Our technique is based on photon-induced nearfield electron microscopy (PINEM)[30-33], an approach in which free electrons exchange energy with the nearfield of the nanostructure in units of light quanta. Usually, PINEM is achieved by the synchronized interaction between electron pulses and laser pulses[30,32,34,35], which can provide nm spatial and fs temporal resolution to image a variety of physical phenomena[10,36-44]. However, the field distribution inside a DLA channel is highly sensitive to the wavelength of the incident laser. A pulse excitation, as is typically used in PINEM, simultaneously excites a range of field distributions and cannot distinguish the individual monochromatic responses. Only few PINEM-type experiments have so far achieved strong enough electron-light interaction without pulsed

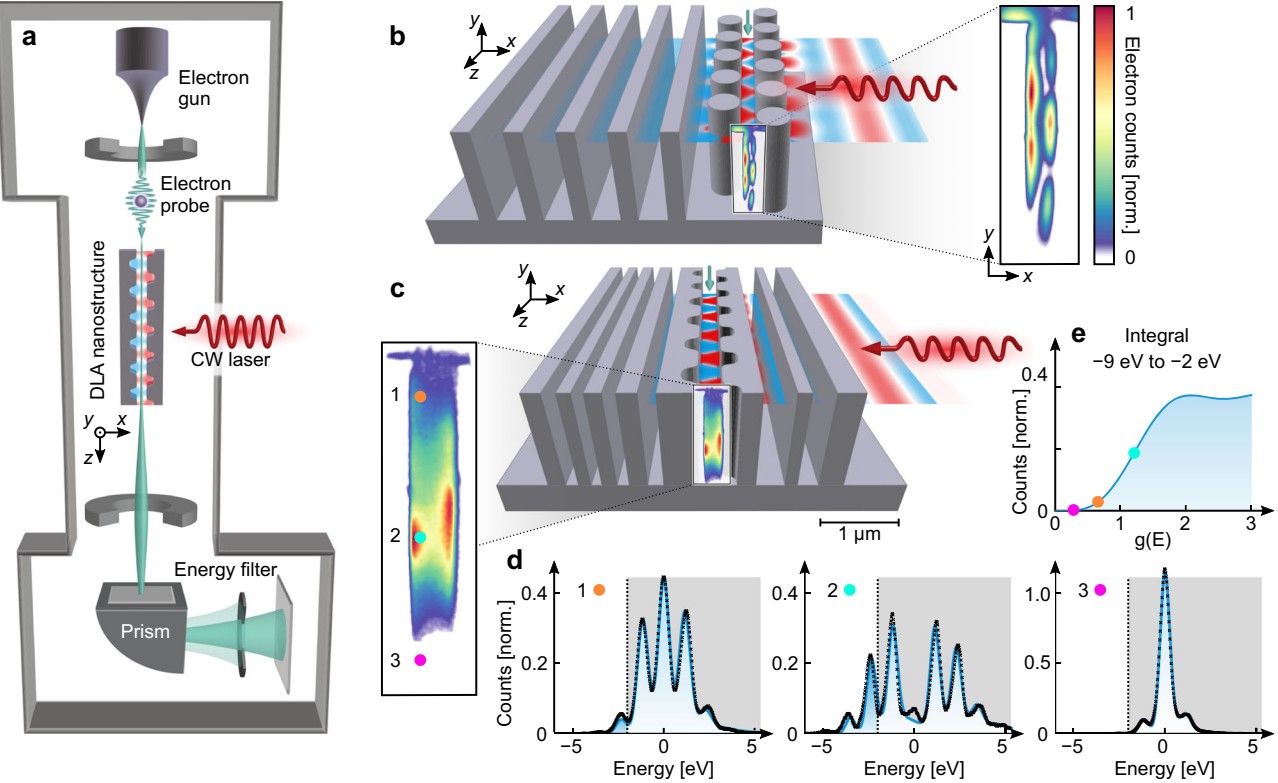

**Fig. 1 | Imaging the field distribution inside dielectric laser acceleration (DLA) structures. a** An illustration of the transmission electron microscope experimental setup. A LaB$_6$ electron source operates in thermionic mode; a set of magnetic lenses and apertures are used to condition the electron beam such that it enters the DLA parallel to the channel at the required spot size; a 1064 nm CW laser modulates the energy of the electrons passing through the 20 µm long DLA structure; the electrons continue to a spectrometer, where they are energy-filtered such that only electrons that gained energy are imaged. **b, c** 3D model of the two structures used in this work: a dual-pillar structure with a distributed Bragg reflector and an inverse-designed resonant structure, respectively. A representative field distribution is overlaid at the end of the channel. **d** Measured (black crosses) and simulated (blue line) electron energy loss spectra (EELS) at representative locations inside the channel of the inverse-designed structure. These energy spectra measurements are obtained with an electron beam spot size of ∼70 nm, which is smaller than the channel widths of 210 and 280 nm in the two DLA structures. In contrast, the electron beam used to image the field distributions (insets of **b** and **c**) has a spot of ∼3 µm, sufficiently large to cover the entire electron channel. The gray region in the spectra was filtered out to obtain the images of the acceleration field profile.

Since the driving field amplitude is about three orders of magnitude smaller than that of fs pulses, the peak energy gain inside the CW-driven DLA structures reaches 5 eV ($g = 1.2$), which in the classical picture corresponds to an acceleration gradient of about 0.2 MeV/m along the effective interaction length of 15 µm, given by the laser spot size. Compared to the much higher acceleration gradients of typically used fs laser pulses, the weak CW light field offers three advantages: (1) The weak field guarantees that the electron energy spectrum is not saturated, i.e., the zero-loss peak in the EELS is not fully depleted, which means that the electron counts monotonically increase with electric field strength. Stronger fields also cause transverse motion of the electron, which reduces the spatial resolution. (2) The CW operation enables working with continuous electron beams, which have much higher flux and better electron beam quality, thus providing better image quality. (3) The narrow bandwidth of CW light enables scanning over the wavelength with sub-nm resolution, which is much narrower than the bandwidth of fs pulses, and reveals the fine spectral response of the DLA. **e** Calculated transformation curve between measured EFTEM counts and the acceleration coefficient $g$, which is proportional to $E_z(x,y)$. The values of $g$ obtained from the fits in **d** are marked with colored dots.

interaction[45–50], which is vital for the characterization of the nearfield inside a DLA channel.

By scanning the wavelength of a continuous wave (CW) laser, we extract the spectral response of our nanostructures, acquiring a deep-subwavelength nearfield image at each wavelength with sub-nm spectral resolution. These nearfield images provide deep insight into the photonic properties of the DLA structures. Specifically, using our imaging technique, we compare the two leading designs of silicon-based DLA structures: (1) the established design based on two rows of pillars[17] with a distributed Bragg reflector on one side (Fig. 1b)[26,51], and (2) an inverse-designed structure with an enclosed acceleration channel for resonant enhancement (Fig. 1c)[27,45,52]. We study the spectral response of both devices and find surprising deviations from their designed field distributions. Through numerical simulations, we can relate those deviations to specific fabrication inaccuracies. We conclude with a sensitivity study of the two structures with respect to changes in their structure dimensions, providing deep insights into current nanophotonic accelerator designs and showcasing our imaging technique of nanophotonic nearfields. This study provides crucial insight for the design of future, more ambitious and complex DLA structures.

## Results

Our experimental setup (Fig. 1a) is based on a transmission electron microscope (TEM) in which thermally emitted electrons are accelerated to an energy of 189 keV and enter the DLA structure (more on the system in ref. 45). A 1064 nm CW laser is focused onto the structure. After the interaction with the optical nearfield in the DLA channel, the emerging electrons reach an electron energy loss spectrometer (EELS), which uses a tunable energy filtering slit and magnetic lenses to create an image of those electrons that gained energy. This way, an image of the acceleration profile $|g(x,y)|$ in the channel is generated $|g(x,y)| = |\int_{-\infty}^{\infty} E_z(x,y,z)e^{-i\omega z/v_e}dz|$ where $E_z(x,y,z)$ is the nearfield distribution of the electric field along the electron trajectory, $\omega$ the angular frequency, and $v_e = \beta c \approx 0.69c$ is the electron velocity. The phase term $e^{-i\omega z/v_e}$ thereby means that the electron samples the specific spatial Fourier mode of the field $E_z^q(x,y,z) = E_z^q(x,y)e^{iqz}$ with $q = \omega/v_e$, which is the mode of interest for electron acceleration.

The Fourier mode $q = \omega/v_e$ imaged by our technique is thus determined by the electron velocity and the laser wavelength, which can be tuned to map the spectral response, as shown below. The DLA structure was designed with a periodicity $\Lambda = 733$ nm and optimized for a laser wavelength of $\lambda = 2\pi c/\omega \approx 1064$ nm. For efficient acceleration, the electron needs to be phase matched to the laser light, that is, its velocity needs to fulfill the synchronicity condition $\lambda = \Lambda/\beta$. How strictly the synchronicity condition has to be fulfilled is determined by the interaction length. For our relatively short structure with an interaction length of 15 μm, we can accommodate deviations in the laser wavelength of up to 25 nm and deviations in the electron kinetic energy of up to 10 keV, without completely losing synchronicity (see Supplementary Information Note 5). The relatively uniform illumination allows us to neglect edge and dephasing effects such that the measured acceleration profile is proportional to the field distribution $|E_z^{\omega/v_e}(x,y)|$ of a single unit cell of the DLA structure, scaling linearly with the number of periods. We note that the DLA structures are designed to have their field dominated by the Fourier mode that contributes to the acceleration, that is, $E_z(x,y,z) \approx E_z^{\omega/v_e}(x,y)e^{iz\omega/v_e}$. We henceforth shorten the notation by writing the field without the $\omega/v_e$ superscript.

We probe the field in the channel by two techniques: (1) Using a parallel electron beam with a small spot size of ~70 nm, smaller than the channel width, we measure the electron energy spectrum at various selected locations in the channel (Fig. 1d). This approach could be extended to image the field distribution by scanning over the channel (a smaller electron spot size could be used for better resolution, but at

the expense of lowering the electron flux). (2) We tune the magnetic lenses controlling the electron divergence and spot size of the TEM to create a collimated electron beam that illuminates the DLA channel homogeneously (with a 3 μm spot size). To acquire an image of the accelerating field, we collected only electrons that gained energy, while filtering out both electrons with no interaction (namely, the zero-loss electrons) and the electrons that lost energy in the interaction. To determine the filtering range (gray region in Fig. 1d), we rely on the first technique to find the optimal energy filter that yields maximum image contrast and field linearity. We note that our field imaging technique with CW light operates in the small energy-modulation regime of few eV – in contrast to the three orders of magnitude higher acceleration gradients found in fs-laser operated DLA structures[22]. Remaining in the weak acceleration regime has a twofold advantage: Firstly, we stay below the point where the zero-loss peak is fully depleted, and the electric field can be retrieved from the EFTEM image (see invertible regime in Fig. 1e). Secondly, the change in electron velocity is negligible and the electron trajectory remains paraxial rather than changing transversely, enabling to image the accelerating field with deep-subwavelength spatial resolution. We note that to observe net acceleration, the electrons need to be pre-bunched before entering the structure, as in other DLA experiments[28].

To understand how the electron counts depend on the electric field strength inside the structure, we consider the electron energy distribution after passage through the DLA structure, see Eq. (1) in Methods. The electron counts equal the integral of the electron energy distribution over the slit of the electron energy filter (width 7 eV, from −9 to −2 eV for the inverse-designed structure and −8.5 to −1.5 eV for the dual-pillar structure). Figure 1e shows the electron counts as a function of the coupling constant $g$. The ideal slit position is chosen such that the field strengths of interest are below saturation, in the regime with maximum slope for maximum image contrast. To image a wide range of field strengths, one can repeat the measurement for multiple slit positions and combine the information into a single field image.

Scanning electron microscopy (SEM) images of the dual-pillar and inverse-designed structure are shown in Fig. 2. The electrons (marked in blue) enter the structures via an alignment aperture and are modulated in the structure by the optical field of a CW laser impinging on the structure perpendicularly to the electron flow. The laser illumination area, marked as a red surface, has a Gaussian distribution with spot diameter of ~15 μm ($1/e^2$), which is smaller than the structure length of 20 μm. The insets in Fig. 2a, b show the simulated xz-field distribution $E_z(x,z)$ of an ideal structure. We note that along the x-direction, the field distributions are always designed to have substantial non-vanishing values at the center of the channel, where electrons can be guided for acceleration. However, our experimental data reveals that the field inside some of the fabricated structures is far from optimal and can even vanish at the center of the channel.

Figure 2c, d presents typical PINEM images of the two DLA channels that were taken without (left) and with (right) electron energy filtering. The unfiltered images show the uniformly illuminated channels, while the filtered images show the PINEM field distribution inside each DLA channel, which is related to $|E_z(x,y)|$ as shown by our theory. The dual-pillar field distribution (Fig. 2c) exhibits several outstanding features. Along the vertical y-direction, the field distribution shows an oscillatory behavior and extends beyond the top of the structure. Furthermore, the field does not extend all the way down to the bottom of the channel. As discussed later, this behavior can be predicted by accurate 3D numerical simulations and can be attributed to the eigenmode profile of the nanostructure sitting on a silicon substrate. For comparison, inspecting the field distribution of the inverse-designed structure (Fig. 2d) reveals intriguing differences and similarities to the dual-pillar structure. Along the vertical y-direction, the field is again not reaching down to the channel bottom, but in contrast

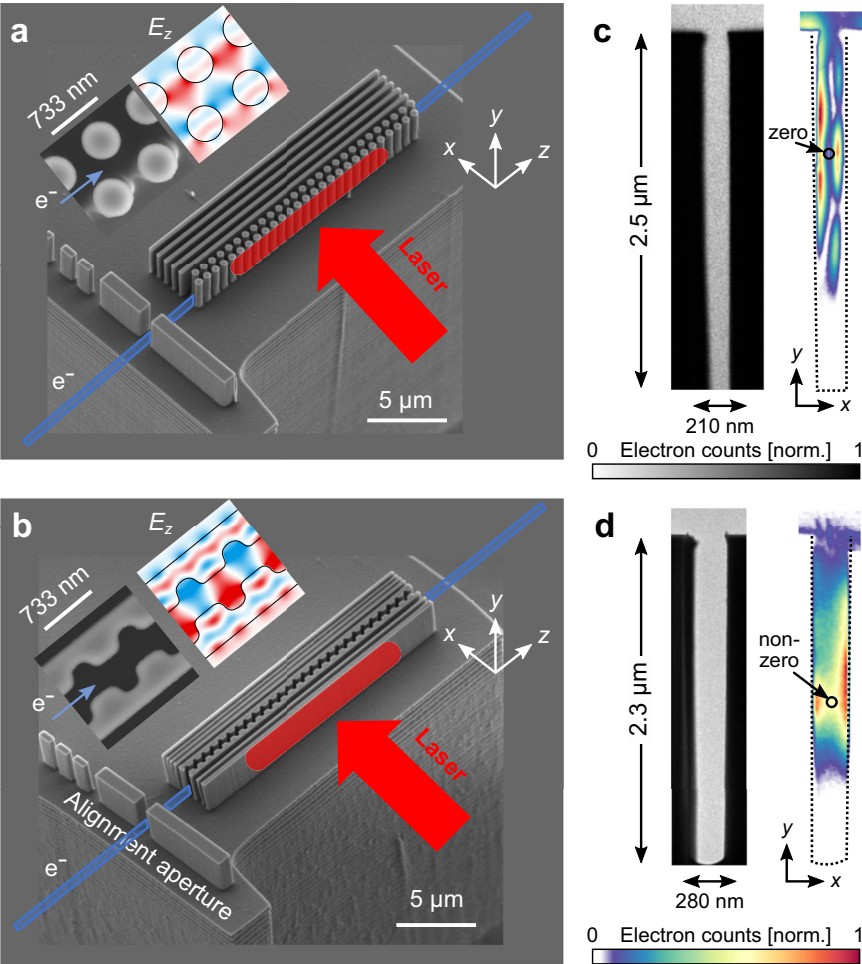

**Fig. 2 | Scanning electron microscopy images of the dielectric laser acceleration (DLA) structures.** The electrons (blue) enter the structures via an alignment aperture and are accelerated in the structure by the electric field of a 1064 nm CW laser impinging on the structure perpendicular to the electron flow. The laser illumination area is marked as a red surface showing that the spot diameter of 15 μm ($1/e^2$) is smaller than the structure length of 20 μm. **a**, **b** The dual-pillar structure with Bragg mirror and the resonance-enhanced inverse-designed structure, respectively. A zoom-in view into the structure and the corresponding accelerating electric field distribution $E_z(x,z)$ are shown as insets for both structures. The periodicity of 733 nm provides phase matching between the electrons and the electric field along the z-direction. Note that $E_z$ has a substantial non-vanishing value at the center of the channel, where electrons are guided. **c**, **d** Typical TEM images of the two DLA channels were taken with a broad 3 μm spot-size electron beam, without (left, black-white color scale) and with (right, rainbow color scale) electron energy filtering. The unfiltered image shows the uniformly illuminated electron channel, revealing a slightly conical shape of the pillars with a slightly narrower channel at the bottom. The filtered image shows the measured PINEM field distribution, which extends above the structure's top and vanishes far above the channel's bottom in both cases. In addition, the field measured in the dual-pillar (**c**) structure has a null at the center, which is different from its design (as discussed below).

to the dual-pillar structure, it does not extend above the channel top, indicating that the field is more strongly confined to the channel. Additionally, the field along the y-direction does not show the oscillatory behavior of the dual-pillar structure.

Along the horizontal x-direction, the measured field of the dual-pillar structure exhibits a null at the channel center (Fig. 2c) – the opposite of the ideal DLA performance. Following Plettner et al.[53], we associate this field distribution with an anti-symmetric mode that follows a hyperbolic sine (sinh) profile. As seen later, the dominance of this mode over the designed symmetric mode that follows a hyperbolic cosine (cosh) distribution can be explained by a fabrication mismatch. For comparison, the field distribution of the inverse-designed structure (Fig. 2d) along the horizontal x-direction, at center height, resembles the expected cosh-like symmetric mode distribution with a non-zero value at the center. A dominant symmetric mode is not only required for optimal acceleration, but when looking forward toward practical applications of DLAs, it can also be used for transverse confinement of the electron beam[28,29].

The field distribution was measured over a range of wavelengths (Fig. 3a, b), from 1063 nm to 1065.4 nm, controlled by changing the CW laser operating temperature (Supplementary Fig. S1). Interestingly, the field distribution of the dual-pillar structure hardly changes with wavelength (Fig. 3a). However, for the inverse-designed structure, the field distribution changes dramatically, with its peak moving from one side to the other.

Figure 3c, d shows that both measured symmetric and anti-symmetric field profiles match with theory[53] and can be approximated accurately with a combination of cosh and sinh functions. When comparing the field distribution to a PINEM image, one further has to consider blurring effects that arise mainly due to the imaging point spread function that we model by a Gaussian convolution. Combining the Gaussian spread with the sinh and cosh mode profiles takes the form $\exp(x^2/2\sigma^2) * [A\cosh(\kappa x) + B\sinh(\kappa x)]$. The coefficients $A$ and $B$ and the evanescent decay rate $\kappa$ are determined by a fit, while the root-mean-square of the Gaussian was chosen as $\sigma = 10$ nm for optimal fit results. At the edges of the channel, electrons scatter off the silicon

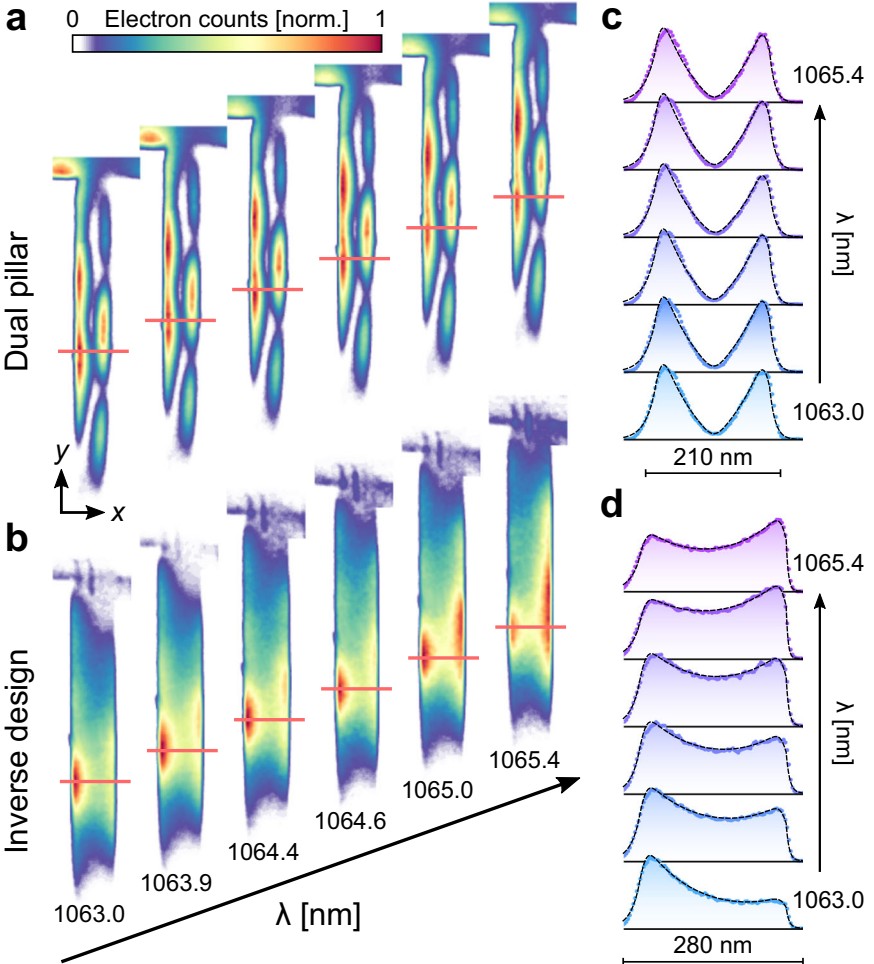

**Fig. 3 | Measured spectral response of the field distribution inside the DLA channel.** Our method for deep-subwavelength spatial imaging with sub-nm spectral resolution is applied to (**a**) the dual-pillar structure and (**b**) the inverse-designed structure. We observe a substantial change for the inverse-designed structure and hardly any change for the dual-pillar structure. **c**, **d** Measured field cross section (blue/purple dots) along the red marking lines in **a** and **b**. The black dashed lines (which can hardly be distinguished from the blue/purple dots since they almost perfectly overlap) are fits of the expected cosh/sinh field profile within the channel.

They were multiplied by an exponential decay at the edges of the channel, where electrons lose energy due to inelastic scattering off the structure. A Gaussian convolution imitates the blurring effect mainly due to the imaging point spread function. The dual-pillar structure has a large sinh component in the fit, which corresponds to an anti-symmetric (odd) structural mode with zero field at the center. In contrast, the inverse-designed structure has a substantial cosh component in the fit, with large non-vanishing energy at the channel center corresponding to a low-order symmetric (even) mode.

boundaries and are either deflected, absorbed, or lose energy through inelastic scattering (e.g., phonon and plasmon emission). These scattering events result in a drop of signal at the boundaries of our PINEM images, which we phenomenologically model by a fitted exponential decay.

Our model (filled dashed curves in Fig. 3c, d) yields excellent agreement with the experimental data (blue/purple dots in Fig. 3c, d) over the entire spectral range that was measured. The measured results show that the field distribution in the dual-pillar structure corresponds almost purely to the anti-symmetric mode with a large sinh contribution in the fit ($A/B \approx 1/10$), whereas the field in the inverse-designed structure corresponds to the symmetric mode distribution with a dominant cosh component in the fit ($A/B \approx 50$). As seen next, the deviation of the dual-pillar field from the design can be explained by a change in the actual pillar diameter relative to the intended design.

To understand the observed field profiles, we performed extensive 3D numerical simulations to study the sensitivity of the field distribution to deviations in the structure dimensions. Figure 4a, b shows the field distributions inside the two structures for small variations of the structure diameters. We singled out the structure diameter $D$

(left column in Fig. 4) as the parameter which best describes the fabrication inaccuracy originating from a too long or too short e-beam illumination time during the lithography process. The best match with the experimental data is obtained for $\delta D = -48$ nm (where $\delta D$ is the change in $D$) for the dual-pillar structure (that is, a 48 nm narrower pillar diameter than the designed diameter of 457 nm) and $\delta D = 0$ nm for the inverse-designed structure.

While both structures show the symmetric mode around the design target ($\delta D = 0 \pm 5$nm), this region is highly sensitive to the structure diameter, and the peak acceleration is seen to shift to one side as the contribution from the anti-symmetric mode increases. This matches our observation from Fig. 3b, where the mode profile of the inverse-designed structure shifted with wavelength. In contrast, our simulations of the dual-pillar structure indicate that the anti-symmetric mode found around $\delta D = -48$ nm is robust to changes in $\delta D$. This robustness agrees with our observation from Fig. 3a, displaying no sensitivity of the dual-pillar field profile to changes in wavelength. Altogether, the inverse-designed structure is far more robust to changes in the structure dimensions because its symmetric mode stretches over a much wider range ($+4$ nm $> \delta D > -40$ nm) than that of the dual-pillar design ($+10$ nm $> \delta D > -4$ nm). Our two fabricated

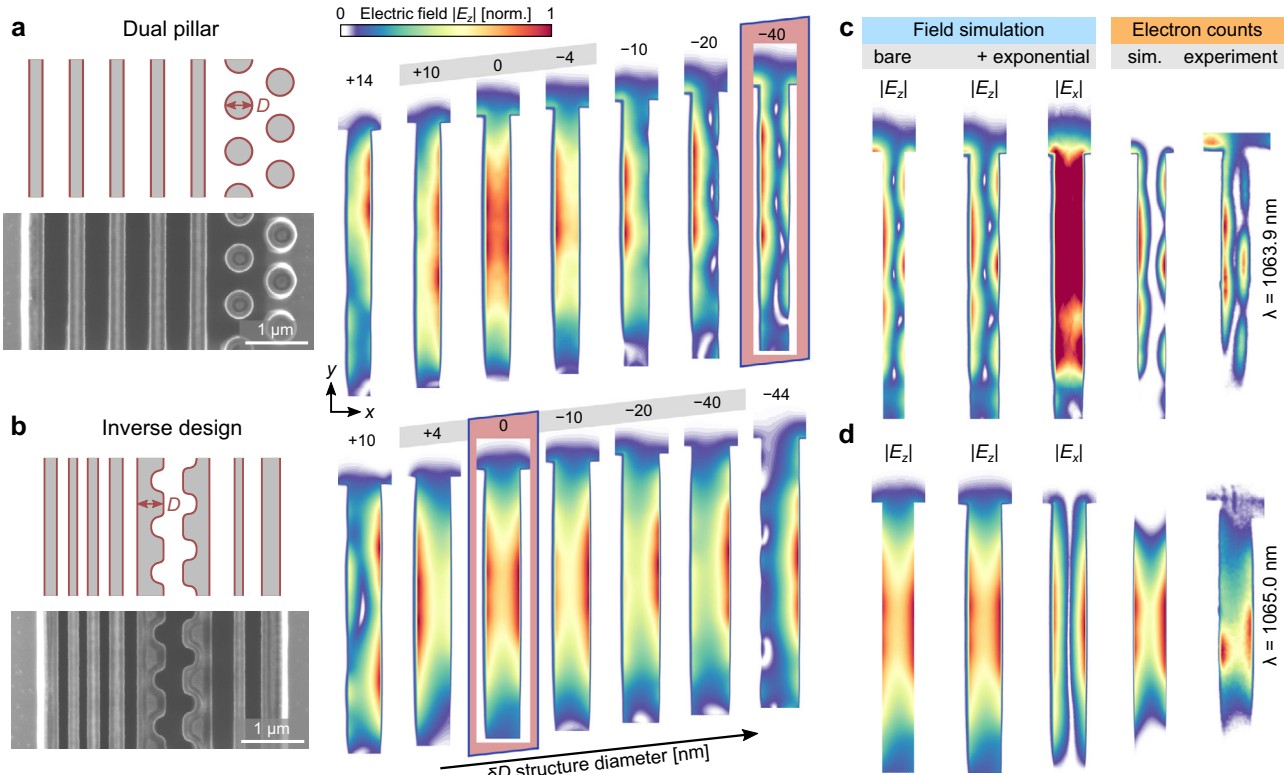

**Fig. 4 | Sensitivity of the DLA nearfield to changes in structure geometry.** The accelerating field distribution $|E_z(x,y)|$ is investigated by 3D simulations as a function of deviations $\delta D$ from the nominal size $D$, marked in the SEM images on the left. **a**, **b** Dual-pillar and inverse-designed structures, showing field simulations for a scan over $\delta D = +14$ to $-40$ nm and $\delta D = +10$ to $-44$ nm, respectively. The best fit results are obtained for $\delta D = -48$ nm and $\delta D = 0$ for the dual-pillar and inverse-designed structures, respectively. Near the design target ($\delta D = 0$), the symmetric mode is dominant for both structures, as designed. For small changes $\delta D$, the field maxima move from side to side with increasing contribution from the anti-symmetric mode. Close to the best fit result of the dual-pillar structure ($\delta D = -40$ nm, marked by red rectangle), its anti-symmetric mode is highly robust against changes in $\delta D$. These simulation results match the experimental findings of Fig. 3, where the symmetric mode of the inverse-designed structure moves from side to side, whereas the anti-symmetric mode of the dual-pillar structure does not change much with wavelength. **c**, **d** Side-by-side comparison of measured and best-fit simulated EFTEM images. For the conversion from electric field to electron counts, we normalized by assuming that the maximum coupling constant $g$ inside the channel reaches 1.8, which is close to the experimentally found value from Fig. 1e. Looking at the deflecting field $|E_x(x,y)|$ reveals that a symmetric mode in $E_z$ is associated with an anti-symmetric mode in $E_x$ and vice versa.

structures are a testimonial of the robustness of the inverse-design approach.

Figure 4c, d demonstrates the amount of knowledge that can be gained when combing our experimental nearfield measurement with 3D simulations. The simulated field $|E_z|$ is shown for the structure parameters that give an optimal replication of the measurement. We added an exponential decay at the vacuum-silicon interface to account for the inelastic scattering of the electrons off the boundaries of the channel. Having found the matching structure parameters this way, we can then extract from the simulation the other field components, $|E_x|$ and $|E_y|$ (see Supplementary Fig. S3). This represents a major advantage over the traditional approach of predicting the field distribution based on measured structure dimensions. Measurements by SEM, for example, can mainly identify the dimensions of outer features of the structure but often misses information inside the nanostructure, since it requires to manually cut the structure at the specific imaging location which is a priory unknown.

The simulated EFTEM images in Fig. 4c, d capture a wide range of details observed in the measurement. Along the horizontal x-direction, we retrieve the anti-symmetric profile for the dual-pillar structure and the symmetric profile for the inverse-designed structure. Along the vertical y-direction, the simulation of the dual-pillar structure reproduced the oscillating pattern as well as the extension of the accelerating field above the left row of pillars. As seen experimentally, the inverse-designed structure does not exhibit this feature above the structure. Lastly, the simulations also

reflect the observation of the field not reaching the channel bottom. These results manifest the need for 3D modeling of these devices, instead of a 2D model that usually approximates the structure as invariant along the vertical direction for convenience and ease of calculations.

The match between experiment and theory allows us to use the simulation for the analysis of the deflecting field $E_x$ (Fig. 4c, d). We found that a symmetric mode in $E_z$ is accompanied by an anti-symmetric mode in $E_x$, and vice versa, which is in accordance with the existing theory[53]. $E_y$ is close to zero throughout most of the channel and only has substantial contribution above the structure (see Supplementary Fig. S3). Thus, not only does the anti-symmetric mode in $E_z$ lead to zero net acceleration, it also strongly deflects the electrons sideways. In contrast, the symmetric mode in $E_z$ has a sinh mode distribution in $E_x$ that can be used to focus the beam along the x-direction, or defocus it, depending on the phase of the field at the time of arrival of the electron[28,29]. A possible application of the anti-symmetric $E_z$ modes in future DLA structures could be for beam deflection. For a 5-μm-short structure and 30 keV electrons, a deflection by 2° inside a 200-nm-wide channel is feasible with existing structures and a previously demonstrated acceleration gradient of 200 MeV/m[26]. For larger deflection angles, the structure itself needs to be curved. The phase of the electron with respect to the laser light determines to which side the electrons are deflected.

We have so far seen that many of the observed features in the nearfield distribution can be attributed to the 3D nature of the DLA

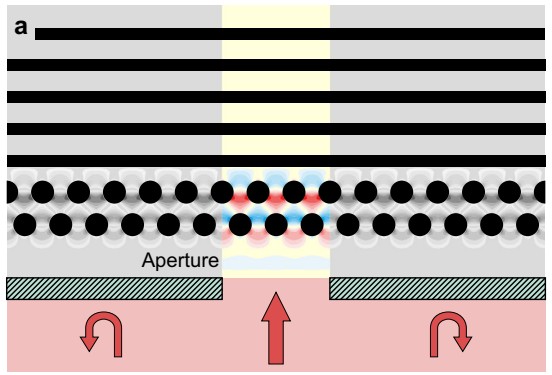
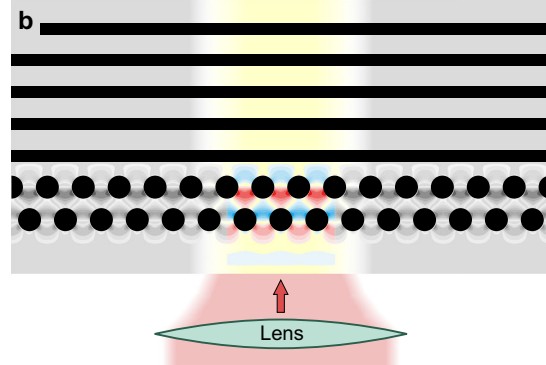

**Fig. 5 | Proposal for 3D tomography of the field distribution inside a DLA structure.** To gain information about how the nearfield behaves along the length of the structure, we suggest illuminating sequentially individual sub-sections, one at a time, to reconstruct the full 3D field. This can be either achieved by (**a**) including an aperture on the chip design. For each position, a dedicated test structure would be added to the chip. **b** Alternatively, a lens can be used to focus the laser to a small spot of a few μm. Scanning the beam along the structure would then provide full 3D field information. A deconvolution algorithm can further increase the spatial resolution along the channel. Simulations of the two approaches are shown in Supplementary Fig. S5.

structure. It is therefore interesting to compare the simulated 2D and 3D results. Supplementary Fig. S2 shows that in the 2D simulation, the inverse-designed structure is expected to have a narrower and stronger spectral response, which can be explained by its resonant enhancement. Notably, the peak acceleration gradient of the inverse-designed structure is higher despite its channel width of 280 nm being larger than that of the dual-pillar structure of 210 nm. In the 3D simulation, the inverse-designed Fabry–Perot-like resonator, which has a height comparable to the wavelength, suffers higher scattering losses, resulting in a drop in the quality factor. This is reflected in a broader spectral response and lower acceleration gradient. These findings highlight the need for 3D simulations to correctly capture the spectral response.

## Discussion

We developed a technique based on continuous-wave PINEM to image the field distribution inside the channels of the two leading DLA structures, achieving deep-subwavelength spatial resolution and showing their spectral response with sub-nm wavelength resolution. While the inverse-designed structure agreed well with its designed symmetric (cosh) field distribution along the horizontal direction, the dual-pillar structure strongly deviated from it and showed a dominant anti-symmetric (sinh) field distribution. By a 3D numerical analysis, we replicated the measured field distributions along both the horizontal and vertical directions and retrieved all three field components in real space. These simulations then enabled us to attribute the measured deviations to a specific fabrication mismatch of the structure diameter (being 48 nm thinner than designed), which shows the precision and prospects of our experimental technique. Our numerical study and experimental examples highlighted the robustness of the inverse-designed structure to changes in the structure dimensions, which is a key advantage in future DLA experiments.

For complex DLA designs with varying periodicity along the electron propagation[18,28,29], we envision extending our approach to 3D tomography of the field inside the device. To this end, we propose to characterize the field at individual cross-sections along the device by selectively illuminating one sub-section at a time. Only the illuminated sub-section will take part in the acceleration process, and we can extract the local nearfield in that sub-section from the measurement. One way to achieve this could be integrating apertures onto the silicon structures. These test structures would sit on the same silicon wafer as the actual DLA structure but would only be used for 3D field characterization. An alternative method is to use a focused laser beam that is scanned along the

structure. Both approaches are schematically illustrated in Fig. 5 and simulated in Supplementary Fig. S5. For further study of the nearfield, one can also perform dark-field imaging to measure the transverse field $E_x$.

Our simulations, backed by our experimental results, shed new light on the predictions about the pros and cons of the two most popular types of DLA structures. While the resonant behavior of the inverse-designed structure originally suggested a stronger acceleration over a narrower bandwidth, we found that it provides a broader bandwidth over a wider range of wavelengths and a wider range of fabrication tolerances. The maximum acceleration gradient at the design parameters is comparable to that of the dual-pillar structure, despite the much broader channel of the inverse-designed structure. For further improvement, a full-3D inverse-design optimization is needed[27].

Looking ahead, the powerful insight gained by continuous-wave PINEM will be of great benefit in the optimization of future DLA designs. More generally, our method can be applied for deep-subwavelength imaging of the field distribution inside many other complex nanostructures and microstructures based on various electron-photon interactions, such as electron wave function shaping structures[54]. Furthermore, one can perform pulsed-laser and pulsed-electron excitations to investigate the dynamic response of such structures.

## Methods

### Experimental setup and fabrication

The experiments were performed in a JEOL JEM-2100 Plus TEM equipped with a Gatan GIF system and the $LaB_6$ electron filament in thermal emission mode. The high current of a continuous electron beam provides us with a considerably better signal-to-noise ratio than operation with electron pulses. The TEM was modified to couple light into the sample (Fig. 1)[10,42,44].

The experimental challenge was to create an electron beam with a small grazing angle such that it could pass through the narrow, 20-μm-long DLA channel, while still being able to resolve the field with a spatial resolution of 10 nm or less. To this end, the TEM was operated in the converged beam electron diffraction (CBED) mode at low magnification. The electron beam diameter in the focal plane was 3 μm for the PINEM measurements and 70 nm for the point-by-point spectrum analysis with a convergence angle of 0.3 mrad and a 0.6 eV FWHM zero-loss energy width. The nanostructure channel was aligned parallel to the electron trajectory using a double-tilt holder (Mel-Build Hata Holder) with a precession of 0.01°, which corresponds to a spatial resolution of 2 nm along the 20-μm-long structure. A custom cartridge

provided clearance for the optical beam. The energy-filtered TEM (EFTEM) image was obtained by using a 7-eV-wide mechanical filtering slit that is located at the exit plane of the EELS prism (Fig. 1). The slit acts as a band-pass filter letting only electrons through that gained energy in the range from −9 to −2 eV for the inverse-designed structure and −8.5 to −1.5 eV for the dual-pillar structure. The electrons are then imaged on the Gatan US1000 camera using a pixel binning of x4. The integration time was ~40s.

A 100 mW CW-driven distributed feedback (DFB) laser (QLD106p-64D0) emitting nominally at 1060 nm served as a seed for a two-stage Yb fiber amplifier with a 4 nm filter between the two stages. The laser beam was focused with a cylindrical lens to achieve a 15-µm Gaussian spot $(1/e^2)$, attenuated to 65 mW, and aligned perpendicular to the electron flow with a precession of $1°$. The angle of incidence of the laser beam plays an important role because it affects both the phase matching between electron and light as well as the excited resonances inside the structure. The generalized synchronicity condition reads $\lambda = \Lambda(1/\beta - \cos(\theta))$, where $\theta$ is the angle of incidence with respect to the electron beam. For our relatively short interaction length of 15 µm, a slight angle deviation of less than $1°$ is negligible (see Supplementary Information Note 5). The effect of an angle on the excited optical mode has been studied by simulation, and a slight angle of $5°$ with respect to the sample surface was found to improve the match between experiment and simulation (see Supplementary Information Note 2). To tune the laser wavelength, the temperature of the built-in TEC was swept. Supplementary Fig. S1 shows the measured beam spectral distribution for different TEC temperatures. The wavelength tunability is limited by the fiber amplifier to approximately 1063–1065.8 nm.

The $2.5 \pm 0.1$-µm-high nanostructures were fabricated by electron beam lithography (100 kV) and cryogenic reactive-ion etching of $1–5\ \Omega$·cm phosphorus-doped silicon[26]. To provide clearance for the laser beam and the electron beam, a 30-µm-high mesa was formed by etching the surrounding substrate away.

**Data post-processing**

The following steps were applied for post-processing of the PINEM images (see Supplementary Fig. S4):

1. Rotation: The original picture was rotated by quadratic interpolation such that the channel is vertically oriented.
2. Dark line removal: In our Gatan imaging filter, the mechanical filtering slit had a local contamination that created a dark line parallel to the horizontal axis. We removed the line by quadratic interpolation of nearby points along the vertical direction. This step was not necessary for the unfiltered data, for which the slit was fully opened.
3. Cropping: The picture was cropped to the region of interest.
4. Colormap change to highlight the relevant non-zero field distributions.
5. Conversion to electric field distribution: To extract the electric field from the measured EFTEM images, we use the PINEM formula for DLA[10]

$$\rho(u,x,y,g) = \rho_0(u,x,y) * \sum_{l=-\infty}^{\infty} J_l^2(2|g(x,y)|)\,\delta(u - l\hbar\omega_0), \quad (1)$$

where $u$ denotes the electron energy, and $\rho_0$ is the initial energy distribution, which accounts for the incoherent part of the spectrum, that is, the zero-loss peak (ZLP) and its transverse spatial distribution. The ZLP is convolved ($*$) in energy with the coherent part on the right-hand side of Eq. (1), which comprises the Bessel function of the first kind $J_l$ of order $l$ and the Dirac delta function $\delta(u)$. The coupling constant $|g(x,y)|$ is approximately proportional to $E_z(x,z)$, as discussed in the main text.

To calculate the electron counts for a value of $g$, we integrate $\rho(u,g)$ over the slit length of the TEM electron energy filter, e.g., from −9 to −2 eV (see Supplementary Fig. S5b). For normalization, we use the unfiltered image (Fig. 2c, d), corresponding to the integration from $-\infty$ to $+\infty$. Supplementary Fig. S5c shows the transformation of the EFTEM image to a distribution of g factors. The features of the EFTEM image are qualitatively preserved, which is why we have used the original EFTEM images throughout the main text.

## Data availability
The data presented in this work is available from the corresponding authors on request.

## Code availability
The code used in this work is available from the corresponding authors on request.

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

## Acknowledgements

The experiments were performed on the UTEM of the AdQuanta group of I.K., which is installed inside the electron microscopy center (MIKA) of the Department of Materials Science and Engineering at the Technion. We thank the MPL cleanroom staff for continued technical support. I.K. and his students gratefully acknowledge the generous support of Ruth Magid and Robert Magid, who have made it possible to carry out these experiments at the UTEM lab at the Technion. I.K. acknowledges funding from the European Union's Horizon 2020 research and innovation program under grant agreement 851780-ERC-NanoEP, the Israel Science Foundation (Grant 830/19) and the European Union's Horizon 2020 research and innovation programme under grant agreement No 101101048-ERC-POC. P.H. acknowledges funding from Gordon and Betty Moore Foundation Grant 4744 (ACHIP), and ERC Advanced Grant 884217 (AccelOnChip).

## Author contributions

All authors contributed substantially to this work. T.F., R.D., M.Y., Y.A., and T.L.A. performed the experiments. U.H. designed the structures and performed the simulations. P.Y. fabricated the structures. R.D., T.F., R.S., O.E., G.E. and I.K. designed the experiment. The manuscript was prepared by T.F. and U.H. with input from all co-authors. P.H. and I.K. conceived the idea and supervised the work.

## Competing interests

The authors declare no competing interests.
