## [Peer Review File · Nature Communications]

Imaging the field inside nanophotonic acceleratorsREVIEWER COMMENTS

Reviewer #1 (Remarks to the Author):

In this work, Fishman et al. apply energy-filtered imaging in a transmission electron microscope to visualize the optical fields inside a continuous-wave-pumped dielectric laser accelerator (DLA). Photon-induced near-field electron microscopy (PINEM) yields qualitative maps of the accelerating optical mode. Two DLA design approaches, a dual-pillar and an inverse-design resonator, are compared and contrasted with numerical simulations, examining the spatial mode profile and optical field homogeneity. Finally, a tomography concept is proposed to characterize the longitudinal field profile by changing the position of the optical excitation along the electron beam axis.

Many studies have applied electron energy loss (EEG) or gain (EEG) spectroscopy to map optical modes at the nano and microscale. Thus, the method presented in itself is not new, but the detailed spatial information gained on the fields in extended DLA structures is highly relevant and complements electron spectroscopic measurements and simulations. Imaging a DLA at the electron kinetic energies it was designed for is an elegant approach to probe and optimize only the relevant structural properties.

Besides these merits, the article lacks technical details and a quantitative comparison between the measured and simulated fields. Therefore, I cannot support publication in its current state. However, the capability for quantitative characterization of micromachined DLAs and sensitivity to design mismatches are highly relevant and may interest scientists from accelerator science to and electron microscopy.

Once the points below are addressed, the manuscript may be suited for publication in Nature Communication.

The title of the manuscript implies a universal method characterizing optical fields inside nanophotonic devices, which is clearly not the case and does not match the abstract/claims. The approach presented only works in transmission through vacuum channels of larger structures, and electrons typically penetrate only through thin sheets of material. This makes it ideal for DLAs, but incompatible with most other photonic structures. The authors should consider a more specific title, reflecting the focus on DLAs.

The authors claim the development of a novel microscopy approach. However, it is explained later that photon-induced near-field electron microscopy in many variants is well-established. What is new in the imaging approach besides combining PINEM with DLA structures?

The introduction gives typical optical fields found in DLAs on the MeV-GeV/m scale. What accelerating fields are achieved in the current continuous-wave pumped structures? Can this be extracted from the experimental data?

How is the image contrast in electron counts connected to the actual optical fields? While the shapes match the simulations qualitatively, a more quantitative discussion is needed. The spectra in Figure 1 suggest a highly nonlinear image contrast.

Are the DLA structures designed to achieve optimum acceleration at 189 keV electron energy? Would the field characterization be valid if the actual electron energy and design target do not match, e.g., due to the excitation of different modes in the DLA?

What is the role of the relative tilt of the electron beam, DLA structure, and laser excitation? Could this influence the excited mode or shift its phase?

In Figure 2, the vacuum channels of the structure seem to get narrow to the bottom. Figures 3 and 4 show deformed images (non-rectangular) in the center. Are these imperfections in the production process or an artifact of the measurement?

The manuscript compares two DLA designs, but the dual pillar structure had a design mismatch in the production. While this illustrates the importance of direct field imaging over simulations, comparing both approaches based on the presented data is difficult and indirect. It should be made clear that most of the comparative discussion originates from the simulations, not the data.

On p9, an additional exponential decay is mentioned that accounts for inelastic scattering. However, only gain scattered data is evaluated and spontaneous scattering at the interface should not be relevant. What is the origin of these electron losses?

The possibility of deflecting the electron beam with an antisymmetric mode is mentioned. What deflection angles could be practically achieved in an extended DLA structure considering the channel width and length, as well as typical fields? Would this be an actual deflection or lead to a smearing of the beam in the transverse direction? A more detailed discussion would be helpful.

Reviewer #2 (Remarks to the Author):

In this work, the author measured the field distribution inside the nanophotonic accelerator through photon-induced nearfield electron microscopy (PINEM), which provides a frequency-tunable deep-subwavelength imaging of the nearfield inside nanophotonic accelerators. Two kinds of nanophotonic structures were studied, including a dual-pillar structure with distributed Bragg reflector and an inverse-designed resonant structure, and their combined experiments and 3D simulations unveiled surprising deviations from the expected designs. This is a very meaningful achievement. The experiment and data are beautiful, and the manuscript is reasonably organized. I have several concerns before accepting the manuscript for publication.

1. In the experiment, the electrons path through the periodic nanophotonic structure under CW laser excitation, in which the laser excites multiple periods of the nanophotonic structure, and the energy

filtered PINEM image only reflects the total “effective field” that the electrons perceived while pathing through the periodic nanophotonic structure. Therefore, the experimentally retrieved field distribution by the PINEM image is not exactly the real field distribution inside the nanophotonic device. As shown by the simulation in Figure S2, the real field distribution inside the nanophotonic device is periodic.

Actually, when the laser excites different number of periods, the PINEM imaging result could be very different. The experiment and the results are beautiful, but it seems the title of the paper is a little bit misleading and over claimed. The authors should try to modify the title to make it more correct.

2. Since the prolonged duration and extended length of the interaction, does the generalized g parameter proposed in previous work (Nat. Phys. 16, 1123–1131 (2020)) be able to describe the interaction between electrons and photons in this work? Can the authors give a simulated PINEM spectrum of Fig. 1d?

3. As for an ideal DAL, the energy of all the electrons, namely, the ZLP of the EELS spectrum should show an overall shift to the energy gain side. While the ZLP has no change and there are only side bands of PINEM around the ZLP. Therefore, in principle, the nanophotonic structures studied are not real DLA structures. Furthermore, the maximum electron gain is only 5 eV, which is too small for a DLA. Can the experiments in this work really reflect the intrinsic properties of a DLA? Is it possible to design a DLA structure to achieve overall acceleration of the electrons?

4. Why not using the femtosecond laser as in the previous work (Phys. Rev. X 11, 041042 (2021)) to do this similar experiment? Any special reason for the use of CW laser and CW electron beam in this work?

5. For the wavelength dependent measurement on the dual-pillar structure, the range of the wavelength is rather limited to get the conclusion. Is it able to try other wavelength to see the difference?

Reviewer #3 (Remarks to the Author):

The work by Fishman, Haeusler, and co-workers utilizes their laser-coupled electron microscope for an energetic analysis of a dielectric laser accelerator (DLA) with the novelty of high spatial resolution. In doing so, they could observe the cumulative effect of the acceleration on the electrons, which was limited to simulations prior to this work.

This manuscript delivers its message clearly and directly, accompanied by informative illustrations and experimental results. In light of that and the growing interest in devices for strong electron-light interaction, I find this paper particularly adequate for Nature Communications and its diverse readership.

NCOMMS-22-34038-T Fishman: Response to referees

We would like to thank all three referees for their time in reviewing our paper, and for providing thoughtful comments and compelling discussion points. Below we address all queries point-by-point and indicate the relevant revisions we made to the manuscript. Since all referees were overall positive about the manuscript, we hope that they will now recommend acceptance of the revised version for publication in *Nature Communications*.

Our responses are marked in green.

Referee #1:

In this work, Fishman et al. apply energy-filtered imaging in a transmission electron microscope to visualize the optical fields inside a continuous-wave-pumped dielectric laser accelerator (DLA). Photon-induced near-field electron microscopy (PINEM) yields qualitative maps of the accelerating optical mode. Two DLA design approaches, a dual-pillar and an inverse-design resonator, are compared and contrasted with numerical simulations, examining the spatial mode profile and optical field homogeneity. Finally, a tomography concept is proposed to characterize the longitudinal field profile by changing the position of the optical excitation along the electron beam axis.

Many studies have applied electron energy loss (EEG) or gain (EEG) spectroscopy to map optical modes at the nano and microscale. Thus, the method presented in itself is not new, but the detailed spatial information gained on the fields in extended DLA structures is highly relevant and complements electron spectroscopic measurements and simulations. Imaging a DLA at the electron kinetic energies it was designed for is an elegant approach to probe and optimize only the relevant structural properties.

We are grateful for the excellent summary of our work and for highlighting its novelty. Indeed, a few works used the electron energy gain spectrum for mapping optical modes, and our work is the first to map modes of DLA structures, which is a unique situation because the interaction is extended, and the coupling is highly sensitive to the electron kinetic energy for which the structure was designed.

Besides these merits, the article lacks technical details and a quantitative comparison between the measured and simulated fields. Therefore, I cannot support publication in its current state. However, the capability for quantitative characterization of micromachined DLAs and sensitivity to design mismatches are highly relevant and may interest scientists from accelerator science to and electron microscopy.

We thank Referee 1 for their constructive comments and thank them for highlighting the broad relevance of our work. We were happy to address all the comments by the referee. Specifically, we now present a quantitative comparison between the measured and simulated fields. Below, we describe the amendments to our manuscript that we have made to address this remark and all other comments.

Once the points below are addressed, the manuscript may be suited for publication in Nature Communication.

A1) The title of the manuscript implies a universal method characterizing optical fields inside nanophotonic devices, which is clearly not the case and does not match the abstract/claims. The approach presented only works in transmission through vacuum channels of larger structures, and electrons typically penetrate only through thin sheets of material. This makes it ideal for DLAs, but incompatible with most other photonic structures. The authors should consider a more specific title, reflecting the focus on DLAs.

We thank the referee for their comment and have changed the title:

Old title:

“Imaging the field inside nanophotonic devices”

New title:

“Imaging the field inside nanophotonic accelerators”

A2) The authors claim the development of a novel microscopy approach. However, it is explained later that photon-induced near-field electron microscopy in many variants is well-established. What is new in the imaging approach besides combining PINEM with DLA structures?

We thank the referee for raising this question. As the referee correctly points out, some individual technical aspects of our work have previously been shown. However, these aspects are not related to the novelty of our work. The microscopy approach that we developed goes beyond the combination of PINEM with DLA structures. More importantly, the novelty is not just in the approach, but also in the phenomena that we access for the first time: allowing us to look at the nearfield inside a nanophotonic acceleration structure.

Furthermore, we are using **continuous-wave** laser light rather than the pulsed excitation typically used in other PINEM studies. This provides two main advantages: (1) The electron current is significantly larger than in pulsed mode which manifests itself in a much better image resolution and a higher signal-to-noise ratio. (2) The continuous-wave laser has a very narrow bandwidth, which allows us to sweep over the laser wavelength on the sub-nm level, enabling to **measure the spectral response** of nanophotonic acceleration structures for the first time.

In parallel with these advantages, we note that there are also unique challenges. Using CW light results in extremely **low acceleration** values of just few eV. We managed to work in this regime and extract the field distribution (see remark 3). Another challenge is the need for an extremely small grazing angle for the electron beam to pass through the narrow, 20- μm -long acceleration channel. To this end, we developed a method for shaping the electron beam such that it is both **highly collimated and narrow**, achieving a high resolution of around 10 nm (only limited by the point spread function of the camera, which can be improved with higher magnification of the magnetic lenses).

To highlight the novelty in our use of PINEM, we also made the following changes:

Previously:

While most works have necessitated a pulsed interaction to reach a strong enough intensity, a few recent works have shown the PINEM-type interaction with continuous-wave (CW) light^{31,48–52}. Our work demonstrates imaging by CW-driven PINEM, which enables the unique combination of deep-subwavelength spatial resolution with a sub-nm spectral resolution.

New (page 4):

Here, for the first time, CW light is used to image the field inside a photonic nanostructure with unprecedented spectral resolution, while offering a much higher signal-to-noise ratio through a continuous electron current. However, the use of CW light requires strong electron-photon interaction, which we achieve through the long interaction length of a many- μm -long DLA structure. The challenge therein is having an electron beam in grazing-angle conditions such that it can pass through our narrow, 20- μm -long acceleration channel while still resolving the field with a spatial resolution of 10 nm or less. To this end, we shaped our electron beam to be both narrow and highly collimated, enabling the unique combination of deep-subwavelength spatial resolution with the sub-nm spectral resolution of CW light.

A3) The introduction gives typical optical fields found in DLAs on the MeV-GeV/m scale. What accelerating fields are achieved in the current continuous-wave pumped structures? Can this be extracted from the experimental data?

We thank the referee for this question about the acceleration gradient in our imaging technique. This question is highly related to the question provided by Referee 2 (see B4, below). In short, our data shows that the *potential* acceleration gradient is similar to that in typical DLAs. But since we intentionally use driving field amplitudes that are about three orders of magnitude smaller than in typical DLA experiments, what we *measure* are acceleration gradients about three orders of magnitude smaller than in such experiments.

Additional information: The maximum energy gain achieved in our work is about 5 eV (compare Fig. 1d). To relate this to the acceleration gradient in the classical picture, we use the coupling constant g , which reaches a peak value of 1.22 (photon energy 1.165 eV) and was extracted via fit (corresponding to a classical peak acceleration of 2.8 eV). The effective interaction length is given by the laser spot size of about 15 μm . Therefore, the acceleration gradient in our structure reaches 0.2 MeV/m. This value is about three orders of magnitude smaller than in typical DLA experiments, however, we note that this value is consistent with a driving field strength that is itself also three orders of magnitude smaller than in such experiments.

We would like to point out that this comparably small acceleration is intentional for two reasons: Firstly, due to the very low laser powers the number of accelerated electrons is not saturated, i.e., the zero-loss peak in the EELS is not fully depleted, which means that the filtered electron counts that create the EFTEM image monotonically increase with electric field strength.

Secondly, high acceleration (as achieved in fs operation) would result in a non-negligible change in electron velocity and alteration of the phase matching condition. Such alterations could prevent uniform imaging across the entire electron channel of the device. This is particularly relevant to long

DLA structures that have increasing periodicity along the structure to accommodate for the change in electron velocity during acceleration. To image such a long structure, one would need to look at each segment of the structure individually (compare Fig. 5) such that the change of periodicity in each segment is negligible.

The referee makes a valuable remark to put our acceleration gradient into context with the typical acceleration gradients in DLA experiment, which is why we made the following changes to our manuscript:

Previously (Figure 1d caption):

Measured electron energy spectra at representative locations inside the channel of the inverse-designed structure. These energy spectra measurements are obtained with an electron beam spot size of 70 nm, which is smaller than the channel widths of 210 and 280 nm in the two DLA structures. In contrast, the electron beam used to image the field distributions (insets of b and c) has a spot of 3 μm , which is larger than the dimensions of the channels in all directions. The gray region was filtered out to obtain the images of the acceleration field profile.

New (Figure 1d caption):

Measured (black crosses) and simulated (blue line) electron energy loss spectra (EELS) at representative locations inside the channel of the inverse-designed structure. These energy spectra measurements are obtained with an electron beam spot size of ~ 70 nm, which is smaller than the channel widths of 210 and 280 nm in the two DLA structures. In contrast, the electron beam used to image the field distributions (insets of b and c) has a spot of ~ 3 μm , sufficiently large to cover the entire electron channel. The gray region in the spectra was filtered out to obtain the images of the acceleration field profile. Since the driving field amplitude is about three orders of magnitude smaller than that of fs pulses, the peak energy gain inside the CW-driven DLA structures reaches 5 eV ($g = 1.2$), which in the classical picture corresponds to an acceleration gradient of about 0.2 MeV/m along the effective interaction length of 15 μm , given by the laser spot size. Compared to the much higher acceleration gradients of typically used fs laser pulses, the weak CW light field offers three advantages: (1) The weak field guarantees that the electron energy spectrum is not saturated, i.e., the zero-loss peak in the EELS is not fully depleted, which means that the electron counts monotonically increase with electric field strength. Stronger fields also cause transverse motion of the electron, which reduces the spatial resolution. (2) The CW operation enables working with continuous electron beams, which have much higher flux and better electron beam quality, thus providing better image quality. (3) The narrow bandwidth of CW light enables scanning over the wavelength with sub-nm resolution, which is much narrower than the bandwidth of fs pulses, and reveals the fine spectral response of the DLA.

Previously (page 5):

To determine the filtering range (gray region in Fig. 1d), we rely on the first technique to find the optimal energy filter that yields maximal image contrast.

New (page 5):

To determine the filtering range (gray region in Fig. 1d), we rely on the first technique to find the optimal energy filter that yields maximum image contrast and field linearity. We note that our field imaging technique with CW light operates in the small energy-modulation regime of few eV – in contrast to the three orders of magnitude higher acceleration gradients found in fs-laser operated DLA

structures²². Remaining in the weak acceleration regime has a twofold advantage: Firstly, we stay below the point where the zero-loss peak is fully depleted, and the electric field can be retrieved from the EFTEM image (see invertible regime in Fig. 1e). Secondly the change in electron velocity is negligible and the electron trajectory remains paraxial rather than changing transversely, enabling to image the accelerating field with deep-subwavelength spatial resolution. We note that to observe net acceleration, the electrons need to be pre-bunched before entering the structure, as in other DLA experiments²⁸.

A4) How is the image contrast in electron counts connected to the actual optical fields? While the shapes match the simulations qualitatively, a more quantitative discussion is needed. The spectra in Figure 1 suggest a highly nonlinear image contrast.

We thank the referee for correctly pointing out that electron counts and optical fields do not follow a linear relation. We agree that this should have been addressed in our manuscript and have made the following changes:

New (page 5):

To understand how the electron-counts depend on the electric field strength inside the structure, we consider the electron energy distribution after passage through the DLA structure, see equation (1) in Methods. The electron counts equal the integral of the electron energy distribution over the slit of the electron energy filter (width 7 eV, from -9 to -2 eV for the inverse-designed structure and -8.5 to -1.5 eV for the dual-pillar structure). Figure 1e shows the electron counts as a function of the coupling constant g . The ideal slit position is chosen such that the field strengths of interest are below saturation, in the regime with maximum slope for maximum image contrast. To image a wide range of field strengths, one can repeat the measurement for multiple slit positions and combine the information into a single field image.

Figure 1 (updated):

We added Fig. 1e, which relates electron counts to the coupling constant g .

Updated (Figure 1 caption):

(e) Calculated transformation curve between measured EFTEM counts and the acceleration coefficient g , which is proportional to $E_z(x, y)$. The values of g obtained from the fits in (d) are marked with colored dots.

Figure 4 (updated):

We added a direct comparison of the simulated electron counts with the measured electron counts.

New (Figure 4 caption):

(c, d) Side-by-side comparison of measured and best-fit simulated EFTEM images. For the conversion from electric field to electron counts, we normalized by assuming that the maximum coupling constant g inside the channel reaches 1.8, which is close to the experimentally found value from Fig. 1e. Looking at the deflecting field $|E_x(x, y)|$ reveals that a symmetric mode in E_z is associated with an anti-symmetric mode in E_x and vice versa.

New (Methods):

The slit acts as a band-pass filter letting only electrons through that gained energy in the range from -9 to -2 eV for the inverse-designed structure and -8.5 to -1.5 eV for the dual-pillar structure. The electrons are then imaged on the Gatan US1000 camera using a pixel binning of x4. The integration time was ~ 40 s.

New (Methods):

Conversion to electric field distribution: To extract the electric field from the measured EFTEM images, we use the PINEM formula for DLA¹⁰

$$\rho(u, x, y, g) = \rho_0(u, x, y) * \sum_{l=-\infty}^{\infty} J_l^2(2|g(x, y)|) \delta(u - l\hbar\omega_0), \quad (1)$$

where u denotes the electron energy, and ρ_0 is the initial energy distribution, which accounts for the incoherent part of the spectrum, that is, the zero-loss peak (ZLP) and its transverse spatial distribution. The ZLP is convolved (*) in energy with the coherent part on the right-hand side of equation (1), which comprises the Bessel function of the first kind J_l of order l and the Dirac delta function $\delta(u)$. The coupling constant $|g(x, y)|$ is approximately proportional to $E_z(x, z)$, as discussed in the main text. To calculate the electron counts for a value of g , we integrate $\rho(u, g)$ over the slit length of the TEM electron energy filter, e.g., from -9 to -2 eV (see Fig. S5b). For normalization, we use the unfiltered image (Fig. 2c and d), corresponding to the integration from $-\infty$ to $+\infty$. Figure S5c shows the transformation of the EFTEM image to a distribution of g factors. The features of the EFTEM image are qualitatively preserved, which is why we have used the original EFTEM images throughout the main text.

New (page 21):
Figure S5. Transformation of EFTEM image to g -factor distribution. **(a)** Fitted electron energy loss spectrum (EELS) of the zero-loss peak (ZLP). The experimentally observed asymmetry in the spectrum was modeled as a sum of a Gaussian and an 11% contribution from an exponentially shifted Gaussian, i.e., the amount by which the Gaussian was shifted to the right-hand side exponentially decays. **(b)** Electron counts as a function of g , calculated via equation (1). The yellow line shows the electron counts without energy filtering, i.e., integration of the EELS over the entire energy axis. The unfiltered EFTEM image without laser illumination served as the normalization for our image transformation to a g -factor distribution. The red line shows the ZLP integrated over the range -2 to +2 eV, which strictly

speaking includes the ZLP and the first sidebands. The blue line shows the number of electrons that gained energy, i.e., the integration of the EELS from -9 to -2 eV, as used for the EFTEM images. (c) Transformation from electron counts (left image) to g factors (right images). It uses the blue line from (b) to map electron counts to g factors. Small electron numbers (below a chosen threshold) cannot be accurately mapped to g factors and were removed.

A5) Are the DLA structures designed to achieve optimum acceleration at 189 keV electron energy? Would the field characterization be valid if the actual electron energy and design target do not match, e.g., due to the excitation of different modes in the DLA?

We thank the referee for this question about the synchronicity between electron and laser light. We probed the field inside the DLA structure over a range of different parameters. A useful rule of thumb for the possible deviations that will not completely lose synchronicity include up to 25nm in the laser wavelength and up to 10 keV in the electron energy. A summary of considerations for such an estimate follows: For phase matching between electron and laser light, the synchronicity condition $\lambda = \Lambda/\beta$ for the laser wavelength λ , the structure periodicity Λ , and the electron velocity β needs to be fulfilled. The range of laser wavelengths and electron velocities for which the synchronicity is sufficiently fulfilled depends on the interaction length of the structure. Here the effective interaction length is about the size of the laser spots (15um or 20 periods). The phase mismatch between electron and light becomes substantial if the deviation of the phase of the electron at the end of the structure, $20 \text{ periods} \cdot \Lambda/\beta$, from the phase of the light, $\lambda \cdot 20 \text{ periods}$, becomes comparable to λ .

To include this discussion into our manuscript, we made the following changes:

Previously (page 4):

If the electron velocity fulfills the synchronicity condition $\lambda = \Lambda/\beta$ for the laser wavelength λ and the structure periodicity Λ , the signal is strongest.

New (page 4):

The DLA structure was designed with a periodicity $\Lambda = 733 \text{ nm}$ and optimized for a laser wavelength of $\lambda = 2\pi c/\omega \approx 1064 \text{ nm}$. For efficient acceleration, the electron needs to be phase matched with the laser light, that is, its velocity needs to fulfil the synchronicity condition $\lambda = \Lambda/\beta$. How strictly the synchronicity condition has to be fulfilled is determined by the interaction length. For our relatively short structure with an interaction length of $15 \mu\text{m}$, we can accommodate deviations in the laser wavelength of up to 25 nm and deviations in the electron kinetic energy of up to 10 keV, without completely losing synchronicity (see Methods).

New (Methods):

Phase matching and Fourier mode sampling

In our experiment, the electron samples the Fourier mode $q = \omega/v_e$ of the electric field in the z -direction. If we neglect edge effects and assume the structure is uniformly illuminated over its interaction length L , then the Fourier transform of the field inside the structure is that of a square pulse of length L multiplied by a periodic function with structure periodicity Λ . In other words, it is the convolution of the sinc-function $\text{sinc}(qL)$ with a sum over delta-functions at $q = n \cdot 2\pi/\Lambda$, where n is the integer mode number. Here, the electron dominantly couples to the $n = -1$ mode, for which

we find the Fourier transform to be $\tilde{E}(q) = \text{sinc}[(2\pi/\Lambda - q)L]$. This means the Fourier component at the electron sampling frequency $q = \omega/v_e$ is proportional to $\text{sinc}[(2\pi/\Lambda - \omega/v_e)L]$. From this, we can retrieve the synchronicity condition $2\pi/\Lambda = \omega/v_e$, which can be rewritten to $\lambda = \Lambda/\beta$.

The first zero crossing of our signal $\text{sinc}[(2\pi/\Lambda - q)L]$ with $L = 15 \mu\text{m}$ occurs for a wavelength of $\lambda = 1064 \text{ nm}$ at an electron velocity of $\beta = 0.672$ (179 keV) or $\beta = 0.706$ (211 keV), and for an electron velocity of $\beta = 0.69$ at a wavelength of 1037 nm or 1089 nm. This means that for the parameters used in our experiment ($\beta = 0.69$ and $\lambda = 1063 - 1065.4 \text{ nm}$) we do not expect to see a significant drop in signal due to phase mismatch.

A6) What is the role of the relative tilt of the electron beam, DLA structure, and laser excitation? Could this influence the excited mode or shift its phase?

We thank the referee for this interesting point of discussion. A tilt of the electron beam with respect to the DLA structure results in a convolution of the acceleration over different points (in the vertical and horizontal directions) across the vacuum channel, which will reduce the spatial resolution. The DLA structure was aligned parallel to the electron beam with a precession of 0.01° , which corresponds to spatial resolution of 2 nm along the 20- μm -long structure, making the tilt of the electron beam with respect to the DLA structure negligible.

A tilt of the laser beam with respect to the surface of the DLA structure, i.e., non-parallel to the surface, was studied by simulation. It showed that an upward tilt by 5° resulted in a slightly better match between experimental data and simulation and was therefore chosen (see Methods).

A tilt of the laser beam with respect to the electron beam, i.e., non-perpendicular illumination, was not studied. It is known that the phase-matching condition changes according to the synchronicity condition $\lambda = \Lambda (1/\beta - \cos(\theta))$, where λ is the wavelength, Λ is the periodicity, β the electron velocity, and θ the angle of incidence with respect to the electron beam. Following the same argument as in question A5, the phase mismatch becomes critical when the angle deviates by more than 5° . However, due to the optical properties of the structure, more significant is how the field inside the structure changes if the laser light is incident under a non-perpendicular angle. For a resonant structure like the inverse-designed structure, the effect of a tilt is expected to result in a shift in the resonance frequency, similar to the tilt of a Fabry-Perot cavity shifting its resonance frequency. We aligned the laser beam perpendicularly to the electron beam with a precision of 1° and thus the above considerations apply.

To include this information into our manuscript, we made the following changes:

Previously (Methods, page 16):

We aligned the nanostructure channel to the electron trajectory using a double-tilt holder (Mel-Build Hata Holder), whose position relative to the electron beam and the tilt angle is determined with nanometer resolution and a step of 0.1° , respectively, using a custom cartridge to avoid shadowing the optical beam.

[...]

The laser beam was focused with a cylindrical lens to achieve a $15 \mu\text{m}$ Gaussian spot ($1/e^2$) and attenuated to $500 \mu\text{W}$ impinging perpendicular to the electron flow.

New (Methods, page 17):

The nanostructure channel was aligned parallel to the electron trajectory using a double-tilt holder (Mel-Build Hata Holder) with a precession of 0.01° , which corresponds to a spatial resolution of 2 nm along the 20- μm -long structure. A custom cartridge provided clearance for the optical beam.

[...]

The laser beam was focused with a cylindrical lens to achieve a 15- μm Gaussian spot ($1/e^2$), attenuated to 65mW, and aligned perpendicular to the electron flow with a precession of 1° . The angle of the incidence of the laser beam plays an important role because it affects both the phase matching between electron and light as well as the excited resonances inside the structure. The generalized synchronicity condition reads $\lambda = \Lambda(1/\beta - \cos(\theta))$, where θ is the angle of incidence with respect to the electron beam. For our relatively short interaction length of 15 μm , a slight angle deviation of less than 1° is negligible (see Methods section on phase matching). The effect of an angle on the excited optical mode has been studied by simulation, and a slight angle of 5° with respect to the sample surface was found to improve the match between experiment and simulation (see below).

A7) In Figure 2, the vacuum channels of the structure seem to get narrow to the bottom. Figures 3 and 4 show deformed images (non-rectangular) in the center. Are these imperfections in the production process or an artifact of the measurement?

We thank the referee for this question. The referee correctly points out that the fabricated structure in Figure 2 has a slightly narrower channel at the bottom as a result of a non-perfectly vertical etch. Figures 3 and 4 appear bulgy at the centre because of the colourmap having a sharp transition from blue to white near the zero value. The field is strongest at the centre, resulting in a bulgy shape. Additionally, the measurements show fewer counts at the bottom because of the slightly conical shape of the structure and the Gaussian shape of the electron beam profile. We made the following changes to clarify this point:

Previously (caption Fig. 2):

The unfiltered image shows the uniformly illuminated electron channel, while the filtered image shows the PINEM field distribution. The measured field distributions extend above the structure's top and vanish far above the channel's bottom in both cases.

New (caption Fig. 2):

The unfiltered image shows the uniformly illuminated electron channel, revealing a slightly conical shape of the pillars with a slightly narrower channel at the bottom. The filtered image shows the measured PINEM field distribution, which extends above the structure's top and vanishes far above the channel's bottom in both cases.

New (Figure S4 caption):

The colormap in (b) gives the false impression of slightly curved channel walls, which is an artefact of having a larger field strength at central height of the channel.

A8) The manuscript compares two DLA designs, but the dual pillar structure had a design mismatch in the production. While this illustrates the importance of direct field imaging over simulations, comparing both approaches based on the presented data is difficult and indirect. It should be made clear that most of the comparative discussion originates from the simulations, not the data.

We thank the referee for this point and made the following adjustments to highlight that the design comparison is mainly based on simulations. The experimental data in our work does indeed only confirm the accuracy of our simulations for the parameters of the fabricated structures of the two types of DLAs.

Previously (page 3):

Comparing the experimental results to the theoretical predictions brings new insights into the current DLA designs, which is crucial for the design of future, more ambitious and complex DLA structures.

New (page 3):

By numerical simulations, we were able to accurately reproduce the structural parameters that resulted in the measured field images. This match allowed us to further use the simulation to compare the sensitivity of the internal field of the two designs with respect to changes in the structure dimensions and laser beam parameters. From this comparison, we obtained new insights into the current DLA designs, which is crucial for the design of future, more ambitious and complex DLA structures.

Previously (Discussion, page 11):

By a 3D numerical analysis, we replicated the measured field distributions along both the horizontal and vertical directions and retrieved all three field components in real space. We were able to attribute the measured deviations to a specific fabrication mismatch of the structure diameter (being 48 nm thinner than designed), showing the precision and prospects of our experimental technique. Our investigation highlighted the robustness of the inverse-designed structure to changes in the structure dimensions, which is a key advantage in future DLA experiments.

New (Discussion, page 11):

By a 3D numerical analysis, we replicated the measured field distributions along both the horizontal and vertical directions and retrieved all three field components in real space. These simulations then enabled us to attribute the measured deviations to a specific fabrication mismatch of the structure diameter (being 48 nm thinner than designed), which shows the precision and prospects of our experimental technique. Our numerical study and experimental examples highlighted the robustness of the inverse-designed structure to changes in the structure dimensions, which is a key advantage in future DLA experiments.

Previously (page 12):

Our experimental results and their match with simulations-shed new light on the predictions about the pros and cons of the two most famous types of DLA structures.

New (page 12):

Our simulations, backed by our experimental results, shed new light on the predictions about the pros and cons of the two most famous types of DLA structures.

A9) On p9, an additional exponential decay is mentioned that accounts for inelastic scattering. However, only gain scattered data is evaluated and spontaneous scattering at the interface should not be relevant. What is the origin of these electron losses?

We thank the referee for this question. The inelastic scattering near the edges of the channel reduces the energy of electrons in those regions. This affects all electrons, irrespective of whether they gain or lose energy through the interaction with the laser light. Therefore, there is a drop in signal near the edges of the channel.

To highlight this, we made the following changes:

Previously (page 8):

At the edges of the channel, electrons lose energy due to inelastic scattering off the silicon boundaries. This results in a drop of signal at the boundaries in our PINEM images, which we model by a fitted exponential decay.

New (page 8):

At the edges of the channel, electrons scatter off the silicon boundaries and are either deflected, absorbed, or lose energy through inelastic scattering (e.g., phonon and plasmon emission). These scattering events result in a drop of signal at the boundaries of our PINEM images, which we phenomenologically model by a fitted exponential decay.

A10) The possibility of deflecting the electron beam with an antisymmetric mode is mentioned. What deflection angles could be practically achieved in an extended DLA structure considering the channel width and length, as well as typical fields? Would this be an actual deflection or lead to a smearing of the beam in the transverse direction? A more detailed discussion would be helpful.

We thank the referee for this interesting point and added the following paragraph to our manuscript:

Old (page 10):

A possible application of the anti-symmetric E_z modes in future DLA structures could be for beam deflection.

New (page 10):

A possible application of the anti-symmetric E_z modes in future DLA structures could be for beam deflection. For a 5- μm -short structure and 30 keV electrons, a deflection by 2° inside a 200-nm-wide channel is feasible with existing structures and a previously demonstrated acceleration gradient of 200 MeV/m²⁶. For larger deflection angles, the structure itself needs to be curved. The phase of the electron with respect to the laser light determines to which side the electrons are deflected.

Referee #2:

In this work, the author measured the field distribution inside the nanophotonic accelerator through photon-induced nearfield electron microscopy (PINEM), which provides a frequency-tunable deep-subwavelength imaging of the nearfield inside nanophotonic accelerators. Two kinds of nanophotonic structures were studied, including a dual-pillar structure with distributed Bragg reflector and an inverse-designed resonant structure, and their combined experiments and 3D simulations unveiled surprising deviations from the expected designs. This is a very meaningful achievement. The experiment and data are beautiful, and the manuscript is reasonably organized. I have several concerns before accepting the manuscript for publication.

We thank Referee 2 for their supportive review, and in particular for their recognition that our work represents “a very meaningful achievement” for the field of nanophotonic accelerators.

B1) In the experiment, the electrons path through the periodic nanophotonic structure under CW laser excitation, in which the laser excites multiple periods of the nanophotonic structure, and the energy filtered PINEM image only reflects the total “effective field” that the electrons perceived while pathing through the periodic nanophotonic structure. Therefore, the experimentally retrieved field distribution by the PINEM image is not exactly the real field distribution inside the nanophotonic device. As shown by the simulation in Figure S2, the real field distribution inside the nanophotonic device is periodic. Actually, when the laser excites different number of periods, the PINEM imaging result could be very different. The experiment and the results are beautiful, but it seems the title of the paper is a little bit misleading and over claimed. The authors should try to modify the title to make it more correct.

We thank the referee for this point of constructive criticism. The referee is correct in pointing out that we are only resolving the field in two spatial directions. The field along the electron channel is experimentally studied by its Fourier component, which is the accelerating field inside this type of structure. We would like to mention that the vast insight that the measured field gives us about the structure dimensions allows us to fully reconstruct the 3D field distribution by simulation. Furthermore, our proposed tomography measurement (in the Discussion) can experimentally resolve the field along the third dimension.

We would like to mention that for the sake of nanophotonic accelerators only the accelerating, periodic mode is relevant. Under the assumption that the fabricated structure is periodic and that the laser illumination is uniform, only the fundamental Fourier mode contributes to acceleration. The effect of all other modes would average out.

We made the following change to the title:

Old title:

“Imaging the field inside nanophotonic devices”

New title:

“Imaging the field inside nanophotonic accelerators”

B2) Since the prolonged duration and extended length of the interaction, does the generalized g parameter proposed in previous work (Nat. Phys. 16, 1123–1131 (2020)) be able to describe the interaction between electrons and photons in this work? Can the authors give a simulated PINEM spectrum of Fig. 1d?

We thank the referee for their valuable suggestion, which we believe adds quantitative strength to our manuscript. We have updated Figure 1 to include the simulated PINEM spectra in Fig. 1d and added Fig. 1e to relate the measured spectra to their respective g values.

Previously (Figure 1):

New (Figure 1):

New (Figure 1 caption):

(d) Measured (black crosses) and simulated (blue line) electron energy loss spectra (EELS) at representative locations inside the channel of the inverse-designed structure. These energy spectra measurements are obtained with an electron beam spot size of ~ 70 nm, which is smaller than the channel widths of 210 and 280 nm in the two DLA structures. In contrast, the electron beam used to image the field distributions (insets of b and c) has a spot of ~ 3 μm , sufficiently large to cover the entire electron channel. The gray region in the spectra was filtered out to obtain the images of the acceleration field profile. Since the driving field amplitude is about three orders of magnitude smaller than that of fs pulses, the peak energy gain inside the CW-driven DLA structures reaches 5 eV ($g = 1.2$), which in the classical picture corresponds to an acceleration gradient of about 0.2 MeV/m along the effective interaction length of 15 μm , given by the laser spot size. Compared to the much higher acceleration gradients of typically used fs laser pulses, the weak CW light field offers three advantages: (1) The weak field guarantees that the electron energy spectrum is not saturated, i.e., the zero-loss peak in the EELS is not fully depleted, which means that the electron counts monotonically increase with electric field strength. Stronger fields also cause transverse motion of the electron, which reduces the spatial resolution. (2) The CW operation enables working with continuous electron beams, which have much higher flux and better electron beam quality, thus providing better image quality. (3) The narrow bandwidth of CW light enables scanning over the wavelength with sub-nm resolution, which is much narrower than the bandwidth of fs pulses and reveals the fine spectral response of the DLA.

(e) Calculated transformation curve between measured EFTEM counts and the acceleration coefficient g , which is proportional to $E_z(x, y)$. The values of g obtained from the fits in (d) are marked with colored dots. The values of g obtained from the fits in (d) are marked with colored dots.

B3) As for an ideal DAL, the energy of all the electrons, namely, the ZLP of the EELS spectrum should show an overall shift to the energy gain side. While the ZLP has no change and there are only side bands of PINEM around the ZLP. Therefore, in principle, the nanophotonic structures studied are not real DLA structures. Furthermore, the maximum electron gain is only 5 eV, which is too small for a DLA. Can the experiments in this work really reflect the intrinsic properties of a DLA? Is it possible to design a DLA structure to achieve overall acceleration of the electrons?

We thank the referee for their comments and would like to answer them in a point-by-point manner:

1.) The referee questions whether the structure can be considered a DLA structure if the mean energy of all electrons remains unchanged in our experiment and asks how overall acceleration of the electrons is possible. To answer this question, one must distinguish between the structure itself and the experiment in which we are using it. In this case, the structure is a true DLA, but it is operated in conditions that do not result in an overall shift to the gain side. As we explain below, this mode of operation is preferable for extracting the field and thus relevant for the operation in DLA conditions.

The reason why we measure about the same number of electrons being accelerated as decelerated is because we sample the entire phase spectrum between electron and laser light. When the electron is in phase with the field, it is accelerated – when it is out of phase, it is decelerated. To the best of our knowledge, all accelerators that use an oscillating field require the electrons to have the right phase to be accelerated. This is indeed the case in most DLA experiments from the earliest observation

[*Nature* **503**, 91–94 (2013)] to very recent experiments [*Science* **373**, 6561 (2021)]. Therefore, the prerequisite of observing acceleration should be on the electrons and not on the structure.

To achieve the right phase matching between electrons and light and observe net acceleration, the electron needs to be bunched before entering the structure. This can be done with the same type of DLA structure, as was demonstrated in [*PRL* **123**, 264802 (2019) and *PRL* **123**, 264803 (2019)].

In summary, our structure can be used to achieve net acceleration if combined with a pre-bunching stage for the electrons. Our imaging technique works as long as some fraction of the electrons are accelerated (or decelerated) and is therefore independent of whether the electrons experience net acceleration. For these reasons, we believe the use of the term “DLA structure” in our manuscript is justified.

2.) The referee notes that the maximum energy gain is only 5 eV and claims that this is too small for a DLA structure. Further, they ask whether our experiment can reflect the intrinsic properties of a DLA structure. We assume that the referee here refers to the way the DLA structure affect electrons when operated with fs laser pulses.

We would like to note that the intrinsic properties of a DLA structure are encoded in its internal field, which scales linearly with the incident field strength. Therefore, our work directly images how the field would look like if operated with much higher powers. The only difference is the wider bandwidth of a fs laser pulse relative to the narrowband nature of CW light. We studied the role of the bandwidth by tuning the CW laser wavelength.

The goal of our experiment was to image the field inside a DLA structure. This goal is best achieved by low power operation so that the electron velocity only changes in a negligible way. Otherwise, the electrons would change their velocity or move transversely depending on their position within the channel profile, complicating the extraction of the field and smearing the spatial resolution. We note that DLA structures are usually designed such that electrons are guided at a certain position within the channel profile and their entire dynamics depends on the transverse field profile [*Nature* **597**, 498–502 (2021)].

To study mm-long DLA structure, we further proposed a tomography technique (Figure 5), where smaller segments of the structure are analyzed. For this tomography technique to work best, the electrons are pre-accelerated to the velocity for which each individual segment was designed for. This does not represent a major challenge for a TEM, which can accelerate to energies of up to 200 keV, covering the design energies of many DLA structures today (~30–100 keV).

We also added to following text to the revised manuscript:

New (page 5 bottom, compared remark A3):

To determine the filtering range (gray region in Fig. 1d), we rely on the first technique to find the optimal energy filter that yields maximum image contrast and field linearity. We note that our field imaging technique with CW light operates in the small energy-modulation regime of few eV – in contrast to the three orders of magnitude higher acceleration gradients found in fs-laser operated DLA structures²². Remaining in the weak acceleration regime has a twofold advantage: Firstly, we stay

below the point where the zero-loss peak is fully depleted, and the electric field can be retrieved from the EFTEM image (see invertible regime in Fig. 1e). Secondly the change in electron velocity is negligible and the electron trajectory remains paraxial rather than changing transversely, enabling to image the accelerating field with deep-subwavelength spatial resolution. We note that to observe net acceleration, the electrons need to be pre-bunched before entering the structure, as in other DLA experiments²⁸.

B4) Why not using the femtosecond laser as in the previous work (Phys. Rev. X 11, 041042 (2021)) to do this similar experiment? Any special reason for the use of CW laser and CW electron beam in this work?

We thank the referee for their interesting question. There are several advantages for using a CW laser over a fs laser:

- 1.) The fs laser pulse will typically saturate the interaction and cause transverse deflections, making it unusable for imaging the field. Of course, the fs pulse field can be attenuated (by at least 3 orders of magnitude) but then it practically goes back to the field strengths of the CW laser, yet with the lower electron flux of the pulsed operation.
- 2.) The fs operation requires operating with the lower flux of electron pulses, reducing the quality of the results. The flux is lower because we have less than an electron per pulse, making the signal impractical for resolving many of the field features.
- 3.) The CW operation offers sub-nm spectral resolution, much narrower than the bandwidth of the fs laser pulse. The spectral resolution allows us to reconstruct the structure parameters by simulation.

To add some points of discussions about the CW/fs advantages, we made the following changes:

Previously (Figure 1d caption):

Measured electron energy spectra at representative locations inside the channel of the inverse-designed structure. These energy spectra measurements are obtained with an electron beam spot size of 70 nm, which is smaller than the channel widths of 210 and 280 nm in the two DLA structures. In contrast, the electron beam used to image the field distributions (insets of b and c) has a spot of 3 μm , which is larger than the dimensions of the channels in all directions. The gray region was filtered out to obtain the images of the acceleration field profile.

New (Figure 1d caption):

Measured (black crosses) and simulated (blue line) electron energy loss spectra (EELS) at representative locations inside the channel of the inverse-designed structure. These energy spectra measurements are obtained with an electron beam spot size of ~ 70 nm, which is smaller than the channel widths of 210 and 280 nm in the two DLA structures. In contrast, the electron beam used to image the field distributions (insets of b and c) has a spot of ~ 3 μm , sufficiently large to cover the entire electron channel. The gray region in the spectra was filtered out to obtain the images of the acceleration field profile. Since the driving field amplitude is about three orders of magnitude smaller than that of fs pulses, the peak energy gain inside the CW-driven DLA structures reaches 5 eV ($g = 1.2$), which in the classical picture corresponds to an acceleration gradient of about 0.2 MeV/m along the effective interaction length of 15 μm , given by the laser spot size. Compared to the much higher acceleration gradients of typically used fs laser pulses, the weak CW light field offers three advantages:

(1) The weak field guarantees that the electron energy spectrum is not saturated, i.e., the zero-loss peak in the EELS is not fully depleted, which means that the electron counts monotonically increase with electric field strength. Stronger fields also cause transverse motion of the electron, which reduces the spatial resolution. (2) The CW operation enables working with continuous electron beams, which have much higher flux and better electron beam quality, thus providing better image quality. (3) The narrow bandwidth of CW light enables scanning over the wavelength with sub-nm resolution, which is much narrower than the bandwidth of fs pulses and reveals the fine spectral response of the DLA.

B5) For the wavelength dependent measurement on the dual-pillar structure, the range of the wavelength is rather limited to get the conclusion. Is it able to try other wavelength to see the difference?

We thank the referee for their remark. We agree that it would be helpful to explore the behavior of the dual-pillar structure over a wider wavelength range. Unfortunately, we are limited by the tunability of our CW laser and could not explore a wider wavelength range.

New (page 17):

The wavelength tunability is limited by the fiber amplifier to approximately 1063–1065.8 nm.

Referee #3:

The work by Fishman, Haeusler, and co-workers utilizes their laser-coupled electron microscope for an energetic analysis of a dielectric laser accelerator (DLA) with the novelty of high spatial resolution. In doing so, they could observe the cumulative effect of the acceleration on the electrons, which was limited to simulations prior to this work.

This manuscript delivers its message clearly and directly, accompanied by informative illustrations and experimental results. In light of that and the growing interest in devices for strong electron-light interaction, I find this paper particularly adequate for Nature Communications and its diverse readership.

We thank Referee 3 for their supportive review and for the particular recognition of the “novelty of high spatial resolution” that our work represents for the field of nanophotonic accelerators. We completely agree that the “growing interest in devices for strong electron-light interaction” makes the work relevant for a diverse readership. We hope that with the substantial improvement of our revised manuscript, all referees will now support publication in Nature Communications.

REVIEWERS' COMMENTS

Reviewer #1 (Remarks to the Author):

The updated manuscript and rebuttal letter by Fishman et al. address my initial concerns in detail. Now a more quantitative relation between the experimental results and simulations is given, which enables the reader to connect both and makes a convincing case for establishing the presented methods for characterizing dielectric laser accelerators. Besides, additional technical details were provided, and the clarity of the descriptions was greatly enhanced.

Therefore, I strongly favor the publication of the manuscript in Nature Communication in its current state. Due to its recency and prospects for designing optimized DLA structures, it may have a significant impact and find an interested readership with an audience from a broader physics background.

Reviewer #2 (Remarks to the Author):

With the careful revision, the authors have addressed all the concerns and the current manuscript has been greatly improved. Considering the meaningful achievement of this work, now I would like to recommend publication of it in Nature Communications.